# Anti-Hermitian photodetector facilitating efficient subwavelength photon sorting

Soo Jin Kim[1,2], Ju-Hyung Kang[1], Mehmet Mutlu[1], Joonsuk Park[1], Woosung Park[3], Kenneth E. Goodson[3], Robert Sinclair[4], Shanhui Fan[5], Pieter G. Kik[6] & Mark L. Brongersma[1,4]

The ability to split an incident light beam into separate wavelength bands is central to a diverse set of optical applications, including imaging, biosensing, communication, photo-catalysis, and photovoltaics. Entirely new opportunities are currently emerging with the recently demonstrated possibility to spectrally split light at a subwavelength scale with optical antennas. Unfortunately, such small structures offer limited spectral control and are hard to exploit in optoelectronic devices. Here, we overcome both challenges and demonstrate how within a single-layer metafilm one can laterally sort photons of different wavelengths below the free-space diffraction limit and extract a useful photocurrent. This chipscale demonstration of anti-Hermitian coupling between resonant photodetector elements also facilitates near-unity photon-sorting efficiencies, near-unity absorption, and a narrow spectral response (∼ 30 nm) for the different wavelength channels. This work opens up entirely new design paradigms for image sensors and energy harvesting systems in which the active elements both sort and detect photons.

[1] Geballe Laboratory for Advanced Materials, Stanford University, 476 Lomita Mall, Stanford, CA 94305, USA. [2] School of Electrical Engineering, Korea University, Seoul 02841, Republic of Korea. [3] Department of Mechanical Engineering, Stanford University, Stanford, CA 94305-3030, USA. [4] Department of Materials Science and Engineering, Stanford University, Stanford, CA 94305, USA. [5] Department of Electrical Engineering, Stanford University, Stanford, CA 94305, USA. [6] CREOL, The College of Optics and Photonics, University of Central Florida, 4000 Central Florida Blvd, Orlando, FL 32816, USA. Correspondence and requests for materials should be addressed to S.J.K. (email: kimsjku@korea.ac.kr) or to M.L.B. (email: brongersma@stanford.edu)

The ability of optically resonant semiconductor and metallic nanostructures to effectively concentrate fields and enhance absorption and emission of light has been one of the greatest successes of nanophotonics[1]. It is by now well-established how the size, shape, and environment of such structures can be tuned to achieve these desirable effects at any target wavelength of interest in the visible and infrared ranges. The enhancement of light absorption has been applied to control photothermal processes at the nanoscale[2,3] and to increase performance of photodetectors[4], sensors[5], imaging systems[6–8], photocatalysis,[3] and solar cells[9–11]. To make further progress and to increase functionality, we need to devise nanophotonics concepts to also achieve control over the polarization, spectral, and angular dependence of light absorption in nanoscale devices. In larger optical systems this is easily accomplished with the help of relatively bulky polarization filters, prisms, color filters, gratings, and apertures. However, such elements are not easily integrated with chipscale devices and it is thus of great value to assess the fundamental performance limitations of nanophotonics approaches.

Spectral separation of photons has been demonstrated using a variety of nanometallic and semiconductor nanostructures[12–15] and inverse design was successfully employed to realize compact wavelength division (de)multiplexing systems[16]. Even the splitting of optical signals at the subwavelength scale and local enhancement of light absorption has been achieved using closely spaced resonant plasmonic elements and metasurfaces[17–25]. However, the application of subwavelength photon sorting in photodetection systems with a narrow spectral bandwidth has remained elusive. The key reasons for this are associated with practical challenges in connecting closely spaced resonant photodetector elements and fundamental limitations in the quality factor/linewidth of subwavelength resonant optical structures[26] that limit spectral resolution. In this work, we demonstrate the possibility to spectrally sort and detect photons with a spectral separation of just 30 nm and below the diffraction limit. This is accomplished by suppressing the near-field interaction and maximizing the far-field interactions between closely spaced photodetector elements. Here, we illustrate how this can be achieved by capitalizing on insights derived from the physics of anti-Hermitian coupling seen in open quantum systems[20].

## Results

**Description of the photon-sorting photodetector.** The proposed device is schematically illustrated in Fig. 1a. It shows a binary grating composed of 60-nm-wide and 80-nm-wide polycrystalline-silicon (poly-Si) nanobeams that are entrenched in a silver (Ag) film. The beams are spaced at a deep-subwavelength pitch, which avoids the formation of first-order

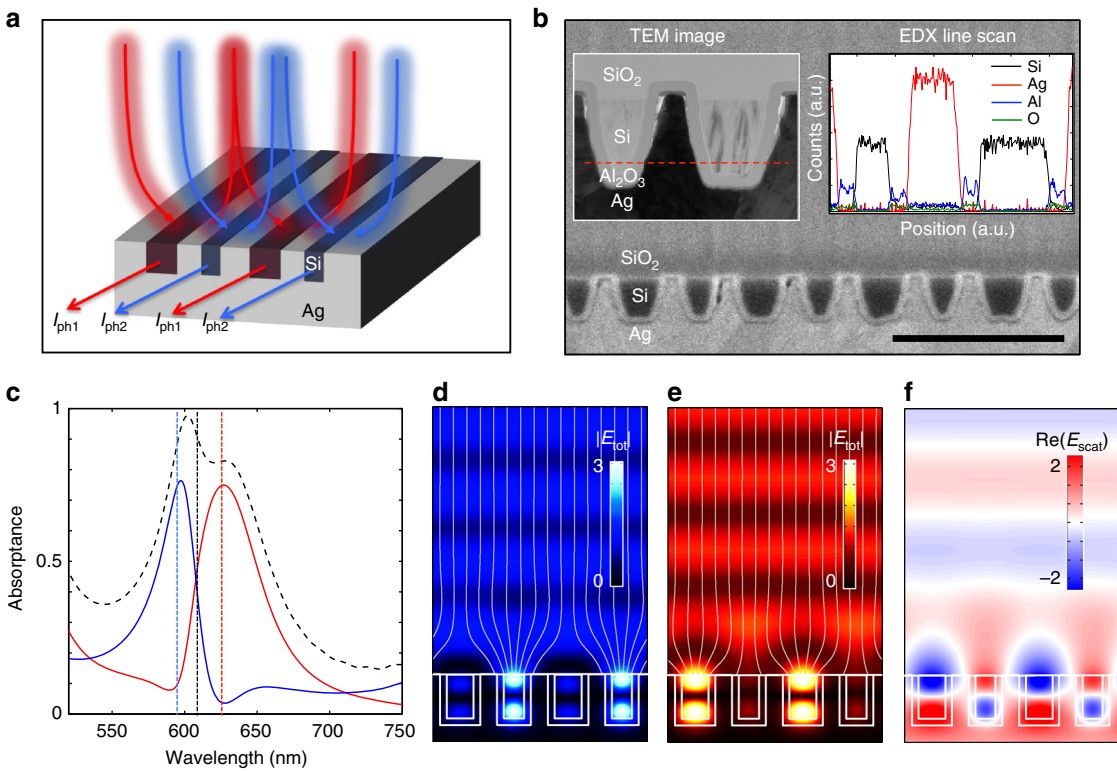

**Fig. 1** A photon sorting, metafilm device that leverages anti-Hermitian coupling. **a** Schematic view of an anti-Hermitian metafilm composed of differently sized semiconductor nanobeams (red and blue). The geometric properties of the nanobeams are chosen to elicit an anti-Hermitian coupling that facilitates sorting of incident photons by wavelength and subsequent photocurrent extraction. **b** Cross-sectional SEM image of a fabricated device structure. Left inset shows the TEM image of a repeating unit cell of the metafilm with electrically insulated Si beams (light) embedded in an Ag film (dark). Right inset shows an EDX line scan along the red dashed line in the TEM image. Scale bar, 500 nm. **c** Simulated absorption spectrums of anti-Hermitian coupled metafilm under TM-polarized illumination with the electric field oriented along the nanobeam axes. The simulated absorption spectrums for the 80-nm-wide (blue curve) and 60-nm-wide nanobeams (red curve) show how the device efficiently sorts photons with a narrow spectral separation of 30 nm. The total absorption in Si material (black dashed line) reaches a near-unity value. Three vertical lines indicate wavelengths of interests at which electric field maps are shown in **d** (blue line), **e** (red line), and **f** (black line). **d**, **e** Simulated images of the total electric field with superposed power flow lines at the wavelengths of 595 nm **d** and 625 nm **e** showing that Mie-like resonances are excited selectively in narrow(/wide) beams at shorter(/longer) illumination wavelengths. **f** Simulated image of scattered electric field at the wavelength of 605 nm showing that the differently sized beams scatter light with a $\pi$ phase difference

diffracted beams upon top-illumination with visible light. As such, the beam array can be treated as a metafilm whose optical properties have been altered from a bulk semiconductor film. A thin, electrically insulating aluminum oxide ($Al_2O_3$) layer separates the Si beams from the Ag film to facilitate effective extraction of photocurrent upon top-illumination. Current can be extracted separately from the sets of wide and narrow beams so that possible differences in their spectral responsivity can be explored.

Figure 1b show a cross-sectional scanning electron microscopy image of a fabricated device. The bi-grating is lithographically defined on a silicon dioxide ($SiO_2$) substrate followed by the deposition of a 17-nm-thick layer of aluminum oxide ($Al_2O_3$) by atomic layer deposition. Finally, an optically thick Ag film is deposited on top of the grating (See Methods). In the experiments, the metafilm is illuminated through the $SiO_2$ substrate. The left inset shows a cross-sectional transmission electron microscopy (TEM) image of one repeating unit cell, showing both a narrow and a wide Si beam. We also performed energy dispersive X-ray spectroscopy along the red dashed line of the TEM image as seen in the right inset. The line scan confirms the presence of a conformally grown $Al_2O_3$ layer capable of electrically separating the Si nanobeams from each other and from the Ag.

Figure 1c shows the simulated absorption spectrums of the metafilm under illumination with transverse magnetic (TM) polarized light, with the electric field oriented along the nanobeam axes. The blue spectrum shows the fraction of absorbed light in the narrow beams and the red spectrum shows the same for the wide beams. From the spectrums, it is clear that 595-nm-wavelength light is most effectively absorbed by the narrow beams and 625-nm-wavelength light is more strongly absorbed in the wide beams. The sum of the absorption in the narrow and wide beams reaches a near-unity value, as shown by the black spectrum. This is useful absorption in the semiconductor that leads to the generation of photocurrent. It is noteworthy that such strong absorption can be achieved in the presence of lossy metals, which tend to cause undesired dissipation. The low losses in the metal are in part due to the fact that the subwavelength grating cannot launch propagating surface plasmon polaritons on the device surface for this polarization.

More intriguing than the very strong overall absorption is the fact that the narrow beams can absorb ~75% of the incident 595 nm light, even though these beams represent just 19% of total surface area. This implies that the absorption cross section $\sigma_{abs}$ of each narrow beam under normal-incidence illumination reaches a value that is about four times larger than the geometric cross section $\sigma_{geo}$, i.e., an absorption efficiency $\eta_{abs} = \sigma_{abs}/\sigma_{geo} \approx 4$. Figure 1d shows how this is physically possible by analyzing the flowlines of the Poynting vector. It shows how normally incident light can preferentially be funneled into the narrow beams that feature a Mie-like optical resonance with two anti-nodes located in the core of the nanobeams[4,27,28]. Note that such resonances are distinct from plasmonic slit resonances, which can be excited when the light is polarized orthogonal to the length of the slit and feature their highest field at the metal/semiconductor interface. At the wavelength of 625 nm in the Fig. 1e, the situation reverses and the wider nanobeams display a resonantly enhanced absorption with a $\eta_{abs} \approx 3$. More subtly, the blue and red absorption spectrums indicate a suppressed absorption in one nanobeam when the other is on resonance. This is highly desirable to reduce any unwanted cross-talk between neighboring pixels that are designed to collect light at distinct wavelengths. Next, we describe how these useful spectral splitting properties arise from the optical coupling between neighboring nanowires in the bi-grating.

**Analysis based on the Anti-Hermitian coupling**. In an effort to understand the possible ways the semiconductor nanobeams can optically couple, previous studies on the light scattering from closely spaced metallic nanoparticle pairs and ensembles provide valuable guidance. For such systems, it has been demonstrated that both near- and far-field coupling can influence the scattering process. An overlap of the excited near-fields of the particles can result in coupled bonding and anti-bonding plasmon oscillator modes that feature resonance frequencies that are shifted from the individual particle resonances[29,30]. Recently, such near-field coupling was also observed for high-index semiconductor nanostructures and linear-combination-of-atomic-orbital style models were developed to describe the observed resonance frequency shifts[31]. Closely spaced metal particles are also known to affect each other's radiation to the far-field and both superradiant and subradiant behavior has been observed depending on the particle spacing[32–36]. Physically, this coupling is mediated by the scattered fields produced by the particles and superradiance and subradiance is achieved when these fields interfere constructively or destructively in the far-field. A recent work[20] has pointed to the interesting analogy that exists between the radiative coupling of metal nanoparticles to the coupling that occurs for open quantum systems consisting of coupled bound and continuum states[37,38]. For such systems, this coupling is described in terms of an anti-Hermitian coupling matrix that links the quasi-bound states. In keeping with this particular viewpoint, we employ temporal coupled mode theory[39,40] (CMT) to derive a matrix equation that describes the optical excitations in a metafilm supporting two coupled resonances in the wide and narrow nanowires. If we assume that these resonances are accessed via a single incidence/exit channel for the light, we can write:

$$\frac{\partial}{\partial t}\begin{bmatrix} c_1 \\ c_2 \end{bmatrix} = i\begin{bmatrix} \omega_1 + i(\gamma_{a1} + \gamma_{r1}) & \omega_{12} + i\gamma_0 \\ \omega_{12} + i\gamma_0 & \omega_2 + i(\gamma_{a2} + \gamma_{r2}) \end{bmatrix}\begin{bmatrix} c_1 \\ c_2 \end{bmatrix} + \begin{bmatrix} \kappa_1 \\ \kappa_2 \end{bmatrix} S_+ \quad (1)$$

where $S_+$ are the incident amplitude normalized such that $|S_+|^2$ denotes the incident power and $c_i$'s are the amplitudes in each modes with $|c_i|^2$ corresponds to the stored energy. $\kappa_i$ quantify the coupling to these modes to external radiation. $\omega_i$, $\gamma_{ai}$ and, $\gamma_{ri}$ in the diagonal terms represent resonance frequencies, absorption loss, and radiation loss, respectively. The off-diagonal terms represents the interaction between the two coupled resonators. Whereas the first off-diagonal term $\omega_{12}$ quantifies the direct near-field coupling, the second part ($\gamma_0$) quantifies the indirect far-field coupling that is dependent on the radiation from the two resonances as $\gamma_0 = \sqrt{\gamma_{r1}\gamma_{r2}}$ (Supplementary Note 1 for the derivation). The relative phase of the radiation fields between the two modes determines whether beams will aid or suppress each other's radiation to the far-field.

Akin to metallic nanoparticles, the considered semiconductor nanobeams support very low radiative quality (Q) factor antenna modes and are thus expected to also feature near-field and far-field, radiative coupling. In our designed device, the metallic fins between semiconductor beams suppress the near-field coupling. As a result, the interaction occurs primarily through far-field coupling. We will illustrate how the far-field coupling of the scattered fields from the narrow and wide beams can be used to spectrally sharpen the absorption properties as compared with individual beams. The most effective far-field coupling is achieved when the beams scatter light into a single optical mode/channel. Figure 1f shows a simulation of the scattered electric field at a selected wavelength of 605 nm, centered between the resonant wavelengths of the wide and narrow beams. It can be seen that the

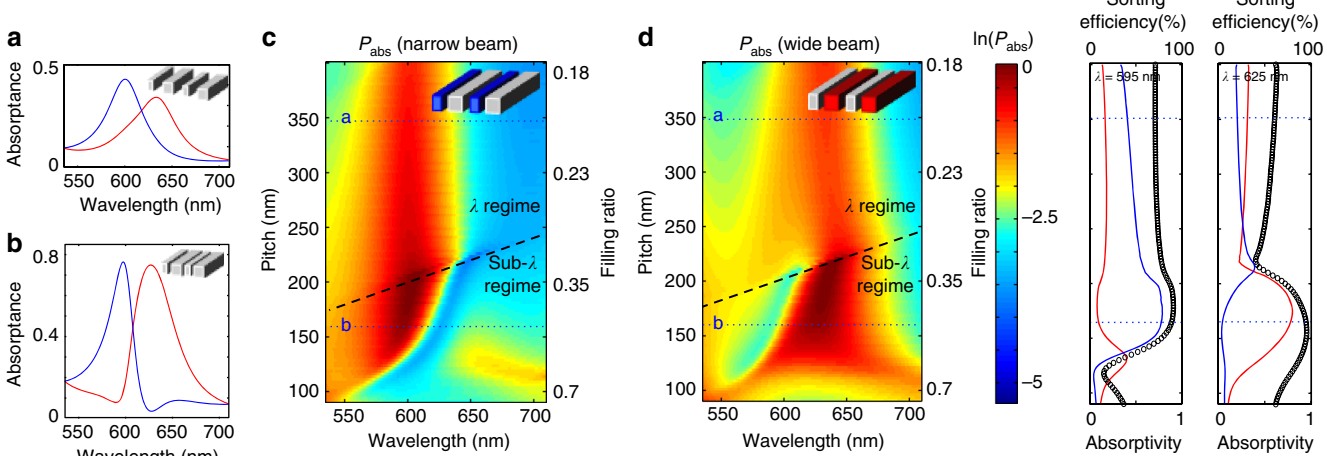

**Fig. 2** Effectiveness of anti-Hermitian coupling in a semiconductor nanobeam array with two beam widths. **a, b** Absorption spectrums of arrays that feature **a** supra- and **b** sub-diffraction limit pitches of 350 and 160 nm, respectively, between adjacent beams. **c, d** Maps of the spectral absorption in just **c** the narrow or **d** the wide nanobeams versus array pitch. The black dashed line indicates the boundary below which first-order diffraction is precluded and a metafilm description becomes valuable. In this regime, pronounced spectral enhancements and suppressions in the absorption are observed. The blue dotted lines indicate the locations at which spectrums shown **a, b** were taken. **e, f** Vertical cross sections of the maps in **c** and **d** taken at the wavelengths of **e** 595 nm and **f** 625 nm, respectively. Blue line shows the absorption in the narrow beams ($P_{narrow}$) and the red line shows the absorption in wide beams ($P_{wide}$). Black circled curve depicts the sorting efficiency defined as the ratio of the absorbed power in the target beam of the total absorbed power $\eta_s$ (%) = [$P_{target}/P_{total}$] × 100, Although the efficiency is over 50% for most periods, it reaches near 100% in metamaterial regime with $P_{total}$ also being close to unity

scattered fields emerging from the differently sized beams feature an approximately $\pi$ phase difference, consistent with the CMT (See Supplementary Note 1). At a small distance from the surface, one can observe flat, horizontal phase fronts of a single outgoing plane wave that results from the coherent addition of the scattered field from the wide and narrow beams. Given the fact that there is only one outgoing plane-wave channel, it is possible to capture the optical properties of the bi-grating with Eq. (1). The nanobeam spacing controls the number of channels by which light can excite/escape the bi-grating structure. As such, this parameter can have a significant impact on the absorption spectrum. The type of changes that can be achieved in terms of the spectral absorption properties of the metafilm are discussed in the next section.

Figure 2a shows how, for a pitch $P = 350$ nm between adjacent nanobeams, the absorption spectrums of the narrow and wide beams feature Lorentzian lineshapes and the spectrums significantly overlap. This pitch is sufficiently large to allow for the generation of first-order diffracted beams. As a result, the interference/coupling of scattered fields from the differently sized beams is weak. The absorption spectrums significantly change when we reduce the pitch to 160 nm, as seen in the Fig. 2b. For such a small pitch, the absorption spectrums feature asymmetric lineshapes and a significantly reduced spectral overlap. At this pitch, no first-order diffraction occurs and the two resonant beams are strongly coupled via the single exit channel. From Eq. (1) it is clear that in the absence of near-field coupling and with a phase difference of $\pi$ in the far-field coupling, the off-diagonal terms in the matrix in Eq. (1) are purely imaginary with both a positive sign. This type of subradiant coupling reduces the radiation leakage (i.e., increases the radiation $Q_{rad}$) and thus increases the local energy storage. Between the resonances, the overall absorption is maximized and close to unity as the scattered fields from the two beams destructively interfere to prevent the generation of a substantial reflected wave. This anti-reflection behavior is further discussed in the description of Supplementary Note 2. It is also important to note that this type

of coupling does not modify the spectral location of the resonance frequencies of the individual beams. On resonance of the narrow beams, the energy storage and thus the light absorption in these beams is maximized. The action of the wide beams is to enhance the energy storage and absorption in the narrow beams by reducing the radiation leakage. This is consistent with the powerflow map shown in Fig. 1d that shows that light enters the slit from a larger absorption cross section[19,28,41]. The absorption cross section of the narrow beams can be so large as to reduce the absorption in the wide beams. This results in a very high sorting efficiency, where virtually all of the incident photons are absorbed in the resonant beam and a very small fraction is absorbed in the off-resonant beam. A similar physical effect emerges in reverse when the wide beam is excited on resonance.

The benefits of subwavelength-scaled devices for spectral sorting are more obviously illustrated in Fig. 2c, d that map the changes in the spectral absorption of the nanobeam array as the pitch is tuned. The absorbed fraction in the narrow nanobeams is shown in Fig. 2c and the absorbed fraction in the wide nanobeams is shown in Fig. 2d. In the regimes below the black dashed lines in both maps, the bi-gratings operate as a metafilm for which the formation of first-order diffracted beams is precluded. This results in effective far-field coupling through a single radiation channel and a more effective sorting of photons into resonant beams. In very deep-subwavelength regime where the pitch between adjacent nanobeams is <100 nm, near-field interaction starts to play a significant role and spectral mode splitting is observed (See Supplementary Note 3).

To quantify how effectively this system is able to spectrally sort light, we define the sorting efficiency ($\eta_s$) as the fraction of absorption at the targeted beam array ($P_{target}$) over the total absorption in silicon ($P_{total}$), i.e., $\eta_s$ (%) = [$P_{target}/P_{total}$] × 100. The sorting efficiency at two closely spaced target wavelengths of 595 and 625 nm are plotted against the pitch in Fig. 2e, f. From these figures, it is clear that the sorting efficiencies reach almost 100% in the metafilm regime. The absolute value of absorbed fraction is ~0.8 for pitches between 150 and 200 nm, as shown by the solid

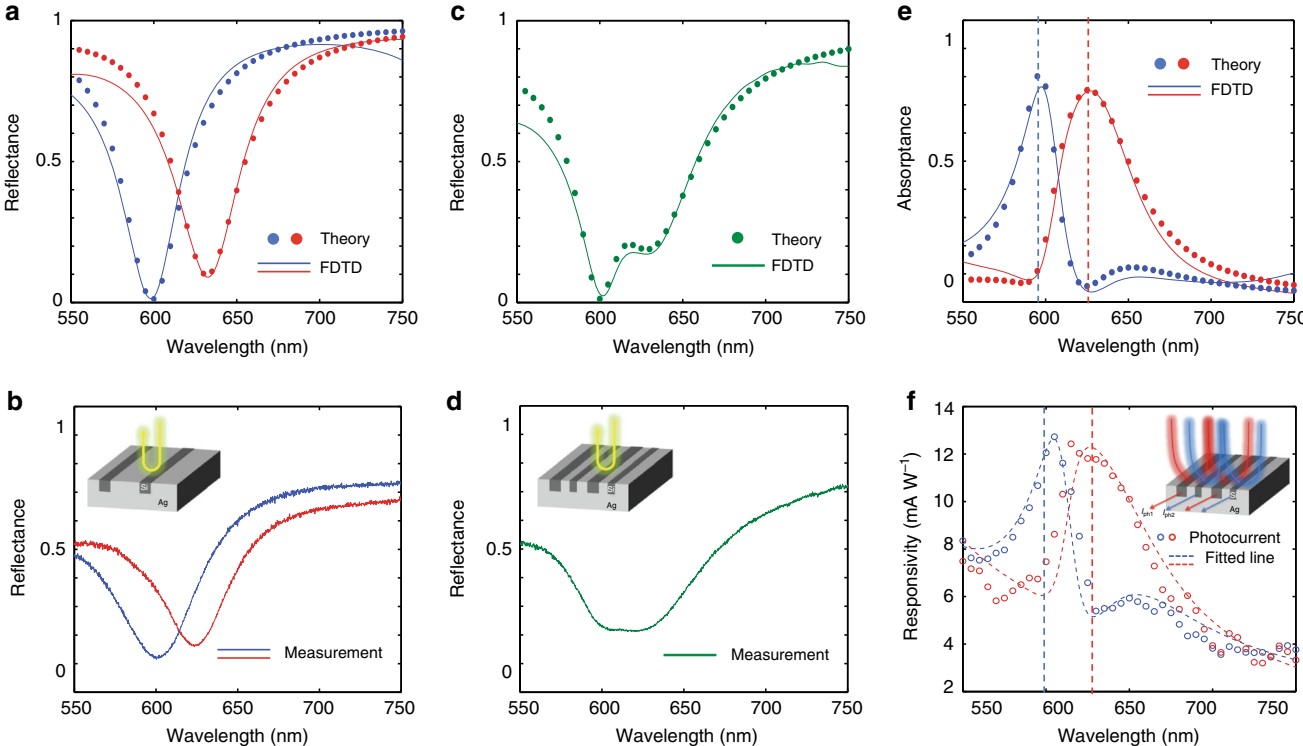

**Fig. 3** Experimental demonstration of spectrally sorted photocurrent generation. **a**, **b** Reflection spectrums of single-sized nanobeam arrays obtained by coupled mode theory (CMT) and finite-difference time-domain (FDTD) simulations **a** and in experiments **b** for beam widths of 60 nm (blue) and 80 nm (red), respectively. **c**, **d** Reflection spectrums of two type of the nanobeams in **a** and **b** interlaced at a subwavelength scale in theory with simulation **c** and in experiments **d**. **e** Absorption spectrum of the multi-sized nanobeam array of **c**, **d** as calculated using CMT and FDTD simulations. Fraction of the absorption is shown in blue for narrow and red for wide nanobeam array. **f** Experimentally extracted photocurrent spectrum depicted as circled point with the fitted line for an anti-Hermitian coupled system. Photocurrent in differently sized nanobeams is extracted separately and measured as seen in blue and red color

red and blue lines. On the other hand, as the pitch size is increased to supra-wavelength values, the achievable sorting efficiencies and overall light absorption significantly decrease.

**Experimental demonstrations.** Last, we experimentally demonstrate the spectral sorting capabilities of bi-grating devices. We start by analyzing two reference devices that feature arrays with single-sized nanobeams that have widths of either 60 or 80 nm. Figure 3a, b show calculated and experimental reflection spectrums from the corresponding devices. The solid line in Fig. 3a shows the simulated reflection spectrum using finite-difference time-domain (FDTD) simulations (Solid line). Each spectrum displays a single reflection dip at a resonant wavelength that is determined by the beam width. This dependence follows from a simple Fabry-Perot style model for groove resonances where trapped light can circulate between the base and exit of the groove that serve as closed and open reflection boundaries, respectively. In such a case, the lowest-order resonance is expected when the free-space wavelength equals $\lambda = 4nd$, where $d$ is the groove depth and $n$ is the mode index of the guided mode supported by the groove. Here, we have to consider the properties of the excited transverse magnetic ($TM_1$) mode with one anti-node of the electric field in the center of the guide. Its mode index increases with increasing width and the resonance for the wider groove is thus redshifted (as opposed to the dependence seen for gap plasmon resonances on grooves width). The spectral dependence of the reflectivity can be described nicely by a CMT for a system supporting a single resonance that is coupled to a single input/output channel (See Supplementary Note 1 for details). From the fits one can then extract useful physical parameters to understand the system operation. For example, for the narrow-beam device

simulation, the best agreement between the FDTD and CMT is obtained for a radiation loss rate $\gamma_r = 6.82 \times 10^{13}$ s$^{-1}$ and an absorption loss rate $\gamma_a = 4.8 \times 10^{13}$ s$^{-1}$. The close match between the radiation and absorption loss rates indicates that the system is near-critical coupling and therefore exhibits near-unity absorption on resonance. The single reflection dip at the target wavelength is also experimentally demonstrated in fabricated narrow-beam and wide-beam devices, as seen in Fig. 3b. Minor discrepancies between experiment and theory, i.e., the experimental non-zero reflection dip with a larger full-width at half-maximum are attributed to slight shape and size variations in the fabricated nanobeams and to non-perfect normal-incident illumination (See Supplementary Note 6 for details).

Figure 3c, d show the simulated and measured spectral reflectance for a bi-grating with 60 and 80 nm beams interlaced. A suppressed reflection can be observed around the resonant wavelengths of the 60 and 80 nm beams. The simulated spectrum as calculated by the FDTD technique can nicely be reproduced using CMT with the same values of $\gamma_r$ and $\gamma_a$ as obtained for the single-sized nanobeams. In addition, we extracted the strength of the indirect far-field coupling $\gamma_0$ between the resonances supported by the two beam sizes (See Supplementary Note 1 for the detailed procedure). Such a model generates scattered fields from the two beams with a phase difference equal to $\pi$ across the resonant dip. The large phase difference between the scattered fields from the beams implies effective anti-Hermitian coupling linked to a destructive far-field interference.

Whereas it is challenging to resolve the individual resonances in a far-field reflection spectrum, photocurrent measurements can access the near-fields in the two beams and demonstrate spectral sorting of light (Fig. 3e, f). Figure 3e shows the absorption in the

narrow and wide nanobeams as simulated by FDTD and as determined by CMT using the same coupling parameters used in Fig. 3c. The experimental spectral dependence of the photocurrent in each of the sets of beams is also plotted in Fig. 3f. Note that photocurrent spectrums were measured separately from the narrow beams and wide beams under the monochromatic illumination with a tunable light source. For each of the two beam widths a clear maximum in the photocurrent is observed near the resonance wavelength for that width. A suppression in the photocurrent is also seen near the resonance wavelength of the wider beam. The observation of asymmetric photocurrent peaks also matches well to the anti-Hermitian model fits shown as red and blue dashed lines. In the experimental case, the overall current increases on the short wavelength-side of the spectrum owing to the stronger intrinsic absorption than predicted by the CMT. The measured peak responsivity is around 12 mA W$^{-1}$ and can be increased through an improved electrical device design, involving electrical doping and improved surface passivation of the NWs. In Supplementary Note 4 we show in theory and experiments that more broadband absorption can be achieved by implementing bi-gratings with a larger difference in the width of the narrow and wide beams. It is also worth noting that this type of spectral sorting is not limited to two sorted wavelengths. By designing metafilms with a larger number of differently sized nanobeams, one can achieve sorting of more wavelengths. Such designs can find potential applications in next-generation solar cells, where the power conversion efficiency can be enhanced through a full utilization of the solar spectrum by directing light of different wavelength ranges into different semiconductor materials with distinct bandgaps. The presented lateral sorting by wavelength is quite distinct from the spectral sorting of light in the stacked layers of multi-junction solar cells.

## Discussion

To summarize, we have demonstrated anti-Hermitian coupling in semiconductor metafilms to design an efficient and integrated photon-sorting detector. By judiciously engineering nanobeam arrays at a deep-subwavelength scale, these metafilm devices can effectively operate similarly to open quantum system with two resonances that interact via the far-field radiation continuum. This affords realization of compact optoelectronic devices that function both as photon sorters and photodetectors in a single, ultrathin layer. Such devices can find application in the field of biosensing, energy harvesting, and optical communication.

## Methods

**Optical simulation**. To optimize and anticipate device performance, optical simulations are performed using the FDTD method. The height of the silicon nanobeam is fixed to 130 nm and the width of nanobeam varies depending on the target wavelengths. Reflectivity is calculated by monitoring reflected power in far-field regime and absorption in silicon nanobeam is selectively calculated by ohmic absorption, i.e., $\omega \cdot Im(\varepsilon) \cdot |E|^2$, where $\varepsilon$ is complex dielectric constant of poly-silicon obtained experimentally from ellipsometry (Woollam). Scattered field is obtained by subtracting incident field from total field, $E_{scat} = E_{tot} - E_{inc}$ and the power flow line is obtained by calculating time-averaged Poynting vector, i.e., $S_{tot} = E_{tot} \times H_{tot}$.

**Device fabrication and measurements**. Semiconductor metafilm is fabricated using standard e-beam lithographic method. First, 130 nm thickness of poly-silicon is deposited on SiO$_2$ substrate by PECVD (Plasma-Enhanced Chemical Vapor Deposition) process. Subsequently, nanobeam array in subwavelength scale is lithographically defined by e-beam process and etching of poly-silicon. Al$_2$O$_3$ layer with the thickness of 17 nm is conformally deposited by ALD (atomic layer deposition) process, and bulk silver layer is deposited by e-beam evaporation. We again deposit a thin Al$_2$O$_3$ layer to prevent the oxidation of silver. Additional lithography and deposition are processed to define aluminum contact layers (See Supplementary Note 5 for details). All the measurements are done by backside illumination through the glass substrate. Reflection spectrums of the fabricated device are measured through a ×20 objective of a confocal optical microscope

(Nikon C1) coupled to a CCD camera and spectrometer (Princeton Instruments). Photocurrent spectrums were measured upon illumination of the devices through a ×10 objective with a supercontinuum laser (Fianium) coupled to AOTF (acoustic-optical tunable filter). Photocurrents from wide and narrow-beam array are extracted separately by preparing two identical samples, each of which are electrically connected to either the narrow or wide-beam array. The illumination source is chopped and the photocurrent is measured using a lock-in amplifier (Stanford Research Systems) and a current meter (Keithley) to attain a high signal-to-noise ratio.

**Data availability**. All data are available from the authors on reasonable request.

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

## Acknowledgements

This work is part of the 'Light-Material Interactions in Energy Conversion' Energy Frontier Research Center funded by the U.S. Department of Energy, Office of Science, Office of Basic Energy Sciences under Award Number DE-SC0001293." S.J.K. was funded by the LMI-EFRC to perform device fabrication, testing, and full-field simulations. We also acknowledge support from the Department of Energy Grant DE-FG07-ER46426 for structural analysis of the devices and analytic optical device models.

## Author contributions

S.J.K. and M.L.B. conceived the idea for the described non-Hermitian photodetector. S.J.K. and J.H.K fabricated and tested the devices. J.P., W.P., and R.S took the TEM images, S.J.K. and M.M. performed the simulations and developed the CMT. All of the authors were involved in analyzing the data and the writing of the manuscript. The projects was supervised by M.L.B.

## Additional information

**Competing interests:** The authors declare no competing financial interests.

