## [Peer Review File · Nature Communications]

Reviewers' comments:

Reviewer #1 (Remarks to the Author):

The authors propose a doubly-resonant structure comprised of Si nanobeams of different dimensions on SiO₂ with an intervening layer of Al₂O₃ and covered with Ag. They show that such a structure can perform as a photodetector that is sensitive to a specific region of the visible spectrum. They connect the design and electromagnetic performance of their structure to coupled mode theory.

The quality of the work is in general very good (theory, experiment, fabrication). I like the connection to coupled mode theory; this is useful to help understand the operating principles and illustrate the underlying physics. The quality of the writing is good but requires a bit of cleanup. The device concept is interesting but not groundbreaking – there are many proposals in the literature for wavelength-selective photodetectors. Also, not much information is given on the photodetector's electronic performance; without this information it is impossible to assess the practicality (and thus the potential impact) of the proposed device. For these reasons, regrettably, I cannot recommend publication.

Specific points:

Comparing the results of Figs. 3a and 3b, the authors attribute the non-zero reflectance dip in part to differences between the modeled and fabricated structures. From SEM cross-sections as shown in Fig. 1b, it is possible to determine the actual size of the structures and then model them precisely. This would seem to be worth the effort. Ditto when comparing Figs. 3c and 3d.

The spectral filtering function of the proposed device is demonstrated by testing narrow and wide nanobeam arrays separately. Thus the spectral sorting function is not fully demonstrated. Can the authors do this on a single device, or at least describe how the structure could be contacted to enable this?

There is essentially nothing reported on the photodetector portion of the device. I think that results on responsivity and dark current are essential. The need for a lock-in amplifier suggests that the detection performance is not very good, in which case it would be important to discuss potential improvements. A description of how the device was probed and biased in the experiments is also needed.

Minor points:

What means "absorptivity" in the ordinate of Fig. 1c? This is a non-standard term.

"Reflectivity" refers to a ratio of E-fields; "reflectance" refers to a ratio of powers. The authors probably mean "reflectance", and if so this should be corrected throughout.

"spectra" is the plural of "spectrums".

I think that the terms "Photon sorting" as used in the paper are inappropriate. These terms suggest sorting of quantum states of light or the detection of a single or a few photons, neither of which are relevant here. Something like "spectral filtering" is probably more appropriate throughout.

Reviewer #2 (Remarks to the Author):

The work is good, but it largely replicates the work done by Crouse et al., in his many works on light and photon sorting gratings and other structures. You really need to cite many of his works in this area, so as to properly give credit to his group for having developed this concept. In particular the works:

Mandel, Isroel, et al. "Theory and design of a novel integrated polarimetric sensor utilizing a light sorting metamaterial grating." *IEEE Sensors Journal* 13.2 (2013): 618-625.

Mandel, Isroel M., et al. "Photon sorting in the near field using subwavelength cavity arrays in the near-infrared." *Applied Physics Letters* 103.25 (2013): 251116.

Jung, Y., et al. "Dual-band photon sorting plasmonic MIM metamaterial sensor." *Proc. SPIE*. Vol. 9070. 2014.

Jung, Young Uk, et al. "Wavelength selective surface plasmons enhanced infrared photodetectors." *Systems, Applications and Technology Conference (LISAT), 2014 IEEE Long Island*. IEEE, 2014.

Mandel, Isroel, et al. "Theory and design of a novel integrated polarimetric sensor utilizing a light sorting metamaterial grating." *IEEE Sensors Journal* 13.2 (2013): 618-625.

Lansey, Eli, et al. "Light localization, photon sorting, and enhanced absorption in subwavelength cavity arrays." *Optics express* 20.22 (2012): 24226-24236.

But there are probably others of Crouse that need to be cited; the authors should research his papers and include all the appropriate ones.

Other than this, the paper is well written, significant, and very interesting. The paper will influence the thinking in this field, the work adds to the work done by Crouse in an original way.

However, with the omission of important historical works on the topic (Crouse's work), I can only recommend the paper for publication once the authors have corrected this critical omission. I can review the resubmitted paper in which these corrections have been made.

Reviewer response:

We thank the reviewers for their comments and valuable assessment of our letter. We feel that we were able to address all of the comments raised by the reviewers and our responses are listed in a point-by-point fashion below.

Reviewer #1 (Remarks to the Author):**Comment 1:**

The authors propose a doubly-resonant structure comprised of Si nanobeams of different dimensions on SiO₂ with an intervening layer of Al₂O₃ and covered with Ag. They show that such a structure can perform as a photodetector that is sensitive to a specific region of the visible spectrum. They connect the design and electromagnetic performance of their structure to coupled mode theory.

Response to comment 1:

We appreciate the reviewer's brief summary of the paper. We address specific points from the reviewer below.

Comment 2:

The quality of the work is in general very good (theory, experiment, fabrication). I like the connection to coupled mode theory; this is useful to help understand the operating principles and illustrate the underlying physics. The quality of the writing is good but requires a bit of cleanup. The device concept is interesting but not groundbreaking – there are many proposals in the literature for wavelength-selective photodetectors. Also, not much information is given on the photodetector's electronic performance; without this information it is impossible to assess the practicality (and thus the potential impact) of the proposed device.

For these reasons, regrettably, I cannot recommend publication.

Response to comment 2:

We thank reviewer for the overall positive evaluation of the paper. We agree with the Reviewer's observation that there are a number of excellent works on wavelength-selective photodetectors. In this work we are for the first time using the physics of Anti-Hermitian coupling to achieve very efficient photon sorting by wavelength in nanoscale photodetectors that have a subwavelength spacing. This has not been achieved before.

The measured peak responsivity of our nanowire detectors is about 12 mW/A, just a few percent of what could be obtained for an ideal detector. This number can be improved through improved electrical device design, involving electrical doping and improved surface passivation of the NWs. The optimization of the electronic performance of nanowire-based photodetectors and solar cells is an active area of research and we believe that with time it will be possible to capitalize on the simulated near-unity absorption predicted for these devices. The optimization of the electrical performance of the presented nanowires would comprise a very significant study in itself that is beyond the scope of this work. We agree with the reviewer that it is important to make a note about the electrical performance and we now show the measured responsivity in Fig. 3f and added some text to discuss this.

Comment 3:

Specific points:

Comparing the results of Figs. 3a and 3b, the authors attribute the non-zero reflectance dip in part to differences between the modeled and fabricated structures. From SEM cross-sections as shown in Fig. 1b, it is possible to determine the actual size of the structures and then model them precisely. This would seem to be worth the effort. Ditto when comparing Figs. 3c and 3d.

Response to comment 3:

We appreciate the reviewer's suggestion. Based on this suggestion, we performed additional simulations based on the SEM images of our fabricated devices that feature non-vertical side walls and slight variations in Si beam size. We added the results in the supplementary part S6. Whereas there are some quantitative differences, the key conclusions of this work remain unaltered.

Comment 4:

The spectral filtering function of the proposed device is demonstrated by testing narrow and wide nanobeam arrays separately. Thus the spectral sorting function is not fully demonstrated. Can the authors do this on a single device, or at least describe how the structure could be contacted to enable this?

Response to comment 4:

We agree the reviewer's point. Although it is demonstrated with two nominally identical devices, a single device can be realized with a more elaborate fabrication strategy. Based on the reviewer's suggestion, we describe the process steps to realize such a device in supplementary information section S5.

Comment 5:

There is essentially nothing reported on the photodetector portion of the device. I think that results on responsivity and dark current are essential. The need for a lock-in amplifier suggests that the detection performance is not very good, in which case it would be important to discuss potential improvements. A description of how the device was probed and biased in the experiments is also needed.

Response to comment 5:

Based on the reviewer's suggestions, we analyzed and added the responsivity of the fabricated photodetector, which replaces Fig. 3f. We discussed this point further in the response to comment 2. We have also added the SEM images of devices in supplementary part S5 to illustrate how the device was probed and biased.

Comment 6:

Minor points:

What means "absorptivity" in the ordinate of Fig. 1c? This is a non-standard term.

"Reflectivity" refers to a ratio of E-fields; "reflectance" refers to a ratio of powers. The authors probably mean "reflectance", and if so this should be corrected throughout.

"spectra" is the plural of "spectrums".

Response to comment 5:

We thank for reviewer's suggestions. Based on the suggestions, we changed the terminology of absorptivity and reflectivity to 'Absorptance' and 'Reflectance' in all axes of our figures.

We also found that we used both terms of ‘spectrums’ and ‘spectra’ in the paper. We unify the terminology as ‘spectrums’ in the paper.

Comment 6:

I think that the terms “Photon sorting” as used in the paper are inappropriate. These terms suggest sorting of quantum states of light or the detection of a single or a few photons, neither of which are relevant here. Something like "spectral filtering" is probably more appropriate throughout.

Response to comment 6:

We thank for reviewer’s suggestions. However, we believe the term “photon sorting” is appropriate in this case. A spectral filter removes certain wavelength components from an incident light wave. A sorting device, sorts light of different wavelengths by redirecting the different wavelength components into different spatial locations/photodetectors. No photons need to be lost in a sorting device and all photons can be used in the photodetection process, in contrast to the case where an absorbing/reflective spectral filter is placed in front of a detector. There are many papers in the literature that use this terminology for structures that sort light waves in essentially classical systems (see e.g.

1. E. Laux *et al.* ‘Plasmonic photon sorters for spectral and polarimetric imaging’, *Nature Photonics* 2, 161 - 164 (2008).
2. I. Mandel *et al.* ‘Theory and design of a novel integrated polarimetric sensor utilizing a light sorting metamaterial grating’, *IEEE Sensors Journal* 13.2, 618 - 625 (2013).

The other referee also notes the relevant work by Crouse on photon sorting metamaterials using the term photon sorting.

Reviewer #2 (Remarks to the Author):

Comment 1:

The work is good, but it largely replicates the work done by Crouse et al., in his many works on light and photon sorting gratings and other structures. You really need to cite many of his works in this area, so as to properly give credit to his group for having developed this concept. In particular the works:

Mandel, Isroel, et al. "Theory and design of a novel integrated polarimetric sensor utilizing a light sorting metamaterial grating." *IEEE Sensors Journal* 13.2 (2013): 618-625.

Mandel, Isroel M., et al. "Photon sorting in the near field using subwavelength cavity arrays in the near-infrared." *Applied Physics Letters* 103.25 (2013): 251116.

Jung, Y., et al. "Dual-band photon sorting plasmonic MIM metamaterial sensor." *Proc. SPIE*. Vol. 9070. 2014.

Jung, Young Uk, et al. "Wavelength selective surface plasmons enhanced infrared photodetectors." *Systems, Applications and Technology Conference (LISAT), 2014 IEEE Long Island*. IEEE, 2014.

Mandel, Isroel, et al. "Theory and design of a novel integrated polarimetric sensor utilizing a light sorting metamaterial grating." *IEEE Sensors Journal* 13.2 (2013): 618-625.

Lansley, Eli, et al. "Light localization, photon sorting, and enhanced absorption in subwavelength cavity arrays." *Optics express* 20.22 (2012): 24226-24236.

But there are probably others of Crouse that need to be cited; the authors should research his papers and include all the appropriate ones.

Response to comment 1:

We thank this Reviewer for making us aware of the important and very relevant work by Crouse et al. We cited and added the most relevant three papers by Crouse et al to the reference list. We believe these references will also put our work in more wide perspective. They complement other early works discussing the sorting of electromagnetic waves that we already cited by Koechlin and Polyakov. (Koechlin, C. *et al.* Total routing and absorption of photons in dual color plasmonic antennas. *Appl. Phys. Lett.* **99**, 38–41 (2011) and Polyakov, A. & Zolotarev, M. Collective behavior of impedance matched plasmonic nanocavities. *Opt. Express* **20**, 470–475 (2012).)

Comment 2:

Other than this, the paper is well written, significant, and very interesting. The paper will influence the thinking in this field, the work adds to the work done by Crouse in an original way. However, with the omission of important historical works on the topic (Crouse's work), I can only recommend the paper for publication once the authors have corrected this critical omission. I can review the resubmitted paper in which these corrections have been made.

Response to comment 1:

We appreciate the reviewer's positive evaluation of our work and note that it add to important works by Crouse in an original way.

REVIEWERS' COMMENTS:

Reviewer #1 (Remarks to the Author):

The authors have satisfactorily addressed all of my comments.

Reviewer #2 (Remarks to the Author):

The revised manuscript is very good. The topic is very important, with a wide range of applications, the technical content of the paper is very good, and the paper is well written and composed. The paper contributes to the knowledge about photon-sorting and will be read and appreciated by the growing number of researchers in this field. The paper should be published in its current form.

Reviewer response:

Reviewer #1 (Remarks to the Author):

Comment:

The authors have satisfactorily addressed all of my comments.

Reviewer #2 (Remarks to the Author):

Comment:

The revised manuscript is very good. The topic is very important, with a wide range of applications, the technical content of the paper is very good, and the paper is well written and composed. The paper contributes to the knowledge about photon-sorting and will be read and appreciated by the growing number of researchers in this field. The paper should be published in its current form.

Response to comments:

We appreciate the reviewers' positive comments and evaluations. All of the reviewers' comments were very helpful to improve the quality of our manuscript.